# Acceptance of Flu Vaccine among Parents of Diabetic Children in Jordan

**DOI:** 10.3390/vaccines12030262

**Published:** 2024-03-01

**Authors:** Walid Al-Qerem, Anan Jarab, Judith Eberhardt, Fawaz Alasmari, Alaa Hammad, Sarah Abu Hour

**Affiliations:** 1Department of Pharmacy, Faculty of Pharmacy, Al-Zaytoonah University of Jordan, Amman 11733, Jordan; alaa.hammad@zuj.edu.jo (A.H.); ph.sarahzakaria2000@gmail.com (S.A.H.); 2Department of Clinical Pharmacy, Faculty of Pharmacy, Jordan University of Science and Technology, Irbid 22110, Jordan; asjarab@just.edu.jo; 3College of Pharmacy, Al Ain University, Abu Dhabi P.O. Box 112612, United Arab Emirates; 4School of Social Sciences, Humanities and Law, Department of Psychology, Teesside University, Borough Road, Middlesbrough TS1 3BX, UK; j.eberhardt@tees.ac.uk; 5Department of Pharmacology and Toxicology, College of Pharmacy, King Saud University, Riyadh 12372, Saudi Arabia; ffalasmari@ksu.edu.sa

**Keywords:** flu vaccine, diabetic children, parental perceptions, vaccine acceptance, Jordan, multinomial logistic regression, healthcare communication, public health education

## Abstract

There is a critical need to understand vaccine decision-making in high-risk groups. This study explored flu vaccine acceptance among Jordanian parents of diabetic children. Employing a cross-sectional approach, 405 parents from multiple healthcare centers across Jordan were recruited through stratified sampling, ensuring a broad representation of socioeconomic backgrounds. A structured questionnaire, distributed both in-person and online, evaluated their knowledge, attitudes, and acceptance of the flu vaccine for their diabetic children. The results indicated that only 6.4% of the study sample reported vaccinating their children against the flu annually, and only 23% are planning to vaccinate their children this year. A multinomial logistic regression analysis revealed notable variability in responses. Specifically, parents with a positive attitude towards the flu vaccine and those with older children had less odds to reject the vaccine (OR = 0.589, 95% CI (0.518–0.670), *p* < 0.001 and OR = 0.846, 95% CI (0.736–0.974), *p* = 0.02, respectively). Conversely, prevalent misconceptions regarding vaccine safety and efficacy emerged as significant barriers to acceptance. Our findings advocate for targeted educational programs that directly address and debunk these specific misconceptions. Additionally, strengthened healthcare communication to provide clear, consistent information about the flu vaccine’s safety and benefits is vital to help enhance vaccine uptake among this vulnerable population, emphasizing the need to address specific concerns and misinformation directly.

## 1. Introduction

Diabetes mellitus (DM) is a widespread metabolic disorder, marked by unusually high levels of glucose in the blood. DM not only contributes to kidney, eye, nerve, and other organ dysfunction but is also considered a significant risk factor for cardiovascular diseases [1]. Type 1 DM represents a chronic pathological state where the patient’s pancreas produces minimal to no insulin [2]. Type 2 DM (previously known as non-insulin dependent type DM) is characterized by insulin resistance, hyperglycemia, and reduced insulin production [3]. The prevalence of DM is increasing at an alarming rate worldwide [3]. As reported by the National Center for Diabetes, Endocrinology, and Genetics (NCDEG), the prevalence of DM in Jordan, when compared to the Middle East and globally, is greater than in any other country [4]. About half of DM patients are undiagnosed and thus more susceptible to DM-induced complications [5]. Approximately 10,000 children and adolescents in Jordan suffer from diabetes [6]. 

Generally, DM patients are more prone to various types of infections, such as respiratory tract, urinary tract, skin, soft tissue, and membrane infections [7]. They exhibit reduced host immunity, which could explain the increased frequency of various infections including influenza (flu) in this group [8]. 

According to the World Health Organization (WHO), seasonal flu infects people regardless of their age. However, its prevalence is greater in children with an average prevalence rate of 20–30% compared to adults with a rate of 5–10% [9]. A significant economic burden is associated with seasonal flu infection due to hospitalizations, deaths, and productivity loss [10]. Seasonal flu is frequently associated with self-limiting, mild symptoms. However, the symptoms tend to be worse in patients with comorbidities and the elderly [11]. Since diabetic patients are more susceptible to hospital admission and other complications associated with flu infection [12], the WHO recommends annual flu vaccination for these patients [13]. Observational studies have reported that vaccinated patients exhibit significantly lower mortality and hospitalization rates compared to unvaccinated patients [14,15,16]. Therefore, although the vaccine may develop several side effects [17], the flu vaccine remains a potential tool for lowering the risk of hospitalization and mortality in patients with chronic diseases affected by flu infection [18]. Despite its proven effectiveness, the vaccination rate is still low and below the target vaccination prevalence rate [19]. A 2019 study found that older adults in Jordan have a negative attitude towards getting the flu vaccine [20]. Common reasons behind individual vaccination refusal are worries about unwanted side effects and disbelief about its effectiveness [21]. Moreover, a cross-sectional study evaluating knowledge and attitudes toward flu vaccination in addition to the vaccination rate among Jordanian adults with chronic diseases, found a low vaccination rate among DM patients [22].

Vaccine safety is the main concern about vaccination acceptance in general [23] and in flu vaccines among parents of diabetic children [24]. Forgetting to get their child vaccinated, familial doubt about vaccine usefulness, refusal by the child, and the negative influence of mainstream media are additional reasons that could explain low flu vaccination rates among diabetic children [25]. 

A study conducted in Singapore to assess flu vaccine knowledge, attitudes, and practices among diabetic patients found that 59.3% of the participants believed that vaccination was an effective tool to prevent influenza and its complications. Although most participants thought that vaccination was effective, only 30.6% had previously received the flu vaccine [26]. Also, in a study conducted in South Africa to evaluate diabetic patients’ knowledge, attitudes, and practices toward seasonal flu and the flu vaccine, most participants felt that the flu vaccine was important for diabetic patients, and 65.4% stated they would recommend it [27]. Moreover, a study conducted in Taif, Saudi Arabia, to assess diabetic patients’ attitudes toward flu vaccination as well as the prevalence of vaccination found that 50.3% of the participants agreed that the flu vaccine was effective, and 51.8% believed that diabetes increased one’s vulnerability to the flu. However, 45.5% thought that the influenza vaccine was dangerous [28]. 

A study focusing on attitudes and beliefs about the flu vaccine among parents of children with chronic medical conditions found that 85.3% felt that their child should receive the flu vaccine, and only 4.8% believed that giving their children the flu vaccine would cause problems [29]. A cross-sectional study in the Middle East revealed that about half of surveyed parents (50.6%) were hesitant about vaccinating their children. This study also showed that 52.5% of the parents of children without chronic illnesses were hesitant about vaccinating their children, whereas only 41% of parents of children with chronic illnesses reported parental vaccination hesitancy [30]. Furthermore, a study carried out with US parents to assess parental hesitancy towards flu and routine childhood vaccination found a higher level of hesitancy among parents of children with poor health toward routine childhood vaccines, but not toward flu vaccines [31].

Since there is no national seasonal flu vaccine program in Jordan, the flu vaccine is not available free of charge to the public and is usually purchased from community pharmacies [32]. The majority (70%) of the Jordanian population is covered by governmental or military health insurance which does not provide free-of-charge seasonal flu vaccines. The remaining 30% are either uninsured or covered by private insurance. Few private insurance plans cover the flu vaccine cost [20].

Diabetic children are particularly susceptible to the adverse effects of the flu, making vaccination an essential protective measure. The lack of focused research on parental attitudes towards flu vaccination for their diabetic children in regions like Jordan and the broader Middle East, presents a significant gap in the public health literature. Thus, this study aimed to assess the acceptance and attitudes of Jordanian parents towards the flu vaccine, thereby contributing to tailored public health strategies and interventions in these underrepresented areas. The study hypothesized that several predictors would impact parental flu vaccine acceptance and practices among parents of diabetic children; these variables include knowledge about the flu, flu vaccine and diabetes, attitude towards flu vaccine, parental sociodemographic status, diabetes status and duration, general health care practices towards diabetes, parents’ and children’s ages, and previous flu vaccination experiences.

## 2. Materials and Methods

This study was conducted with parents of diabetic children in outpatient clinics at two public Jordanian hospitals. The first, King Abdullah University Hospital (KAUH), is situated in northern Jordan and provides extensive medical services to patients from the northern governates of the country. The second, AlBashir Hospital is situated in Amman, the capital of Jordan, and provides medical services to a significant portion of the population from various regions. These factors ensured the representativeness of the sample within the Jordanian context. 

The inclusion criteria encompassed parents of diabetic children aged 18 years or younger who expressed interest in participating in the study. The research pharmacist retrieved the list of patients with appointments at the pediatric endocrine clinic on the same day and identified patients who met the inclusion criteria. The parents of these patients were approached and were provided with a concise description of the aims of the study. All participants were informed about the anonymity and confidentiality of the information collected, as well as the voluntary nature of their participation. Additionally, all participating parents signed an informed consent document. The interview was conducted in a separate room at the outpatient clinic and took an average of 10 min to complete.

A total of 445 parents were approached, of whom 405 (91%) consented to participate in the current study. The data were collected between 17 August 2023 and 5 January 2024. The research was conducted in accordance with the ethical principles stated in the Declaration of Helsinki, receiving ethical clearance from the Jordan Ministry of Health (Reference #MOH/REC/2023/119), Al-Zaytoonah University of Jordan (Reference #22/20/2022–2023), and Jordan University of Science and Technology (Reference #2022/07).

### 2.1. Data collection and Study Instruments

#### 2.1.1. Customized Questionnaire

Data collection was conducted using a custom-designed questionnaire developed on Google Forms, through an extensive literature review to ensure its comprehensiveness and relevance. Subsequently, it was translated from English to Arabic. It was composed of five parts, with the first section dedicated to collecting demographic information. This included details about the parents such as age, gender, education level, socioeconomic status, and income, as well as information about the child including age, gender, duration of diabetes (DM), HbA1c levels, and exposure to second-hand smoke. The second part assessed participants’ knowledge of DM (six items), flu (four items), and the flu vaccine (four items). The knowledge score was computed by granting one point for each correct answer and zero points for incorrect or uncertain responses. The third part contained three items assessing flu vaccination behavior and prior experiences (including any vaccine-associated side effects experienced). The fourth part evaluated parents’ attitudes toward flu vaccination for their diabetic children (eight items). The fifth part assessed the diabetic patients’ self-management, and included four self-care activities (diet, physical activity, blood glucose testing, and foot health). The last section evaluated participants’ willingness to vaccinate their children and the obstacles they faced in getting their children vaccinated. Scores for knowledge and attitudes were derived from the responses to the questions, awarding one point for every accurate response in the knowledge domain, with no points for incorrect ones. The fourth section utilized a Likert-scale response format (ranging from strongly disagree to strongly agree), assigning 5 points for “strongly agree” down to 1 point for “strongly disagree”. Items phrased negatively were scored in reverse.

#### 2.1.2. Questionnaire Validation

Content validity was assessed by a group of specialists, including a clinical pharmacy professor, two endocrinologists, and two clinical pharmacists. The questionnaire was developed based on two previous studies, one conducted among parents of children with asthma and the second among individuals with diabetes, in Jordan [33,34]. To align with the English literature review, the questionnaire was originally developed in English. Subsequently, it was adjusted and customized to align with the study sample and then translated/back-translated into Arabic, the official language of Jordan. The translation process involved two independent translators, leading to two similar versions of the questionnaire. Additionally, a preliminary study with 30 participants was conducted to test the clarity of the questionnaire for the Jordanian audience, although the data from this preliminary phase were excluded from the final analysis. Moreover, Cronbach’s alpha coefficients were computed to assess the reliability and internal consistency of the latent variables, i.e., knowledge of DM, flu, and the flu vaccine, participants’ attitudes towards flu immunization for children with diabetes, and the Diabetes Self-Care Activities Measure (SDSCA). A Cronbach’s alpha score of 0.7 or above is deemed satisfactory. The obtained Cronbach’s alpha scores varied from 0.746 to 0.794, thereby surpassing the threshold of 0.7.

#### 2.1.3. The Validated Arabic Adaptation of the Summary of Diabetes Self-Care Activities Measure (SDSCA) 

Parental self-care behavior in relation to managing their children’s diabetes was assessed using the Summary of Diabetes Self-Care Activities Measure (SDSCA) [35]. This includes 10 items, using an 8-point Likert scale to gauge the frequency of specific diabetes self-care behaviors (such as diet, physical exercise, blood glucose monitoring, and foot care) over the previous 7 days. The participant indicates the number of days that the patient performs the activity. The total score is calculated by averaging scores on all 10 items. 

### 2.2. Sample Size Calculation

Considering a convenience sampling approach, the calculated minimum sample size was 385 with a 95% significance level (α = 0.05, β = 0.2), a 5% margin of error, and a 50% population proportion. A total of 405 parents participated in the study [36].

### 2.3. Statistical Analysis

SPSS software version 28.0 was used to analyze the data. The frequencies and percentages were used to represent the categorical variables, while the median and 25–75 percentiles represented the continuous variables. 

A multinomial logistic regression model was developed to assess the factors influencing the intention to obtain the flu vaccine in the current year. The model incorporated independent variables in the form of age and gender of the parent, along with their educational level, socioeconomic status, the child’s age and gender, DM duration, and HbA1c level. Furthermore, it included knowledge of DM, flu, and flu vaccine, DM self-care practices, attitudes towards the vaccine, and whether the child had previously received the flu vaccine. The model goodness of fit was evaluated by computing Nagelkerke’s R^2^. A *p*-value below 0.05 was deemed to indicate statistical significance.

## 3. Results

The sociodemographic characteristics for the 405 enrolled parents and their diabetic children are displayed in Table 1. The children were aged between 6 and 13; most of them were females (51.9%). The parents were aged between 32 and 43 and were mostly females (62.5%). Most had a high school degree or lower (60.2%), and the vast majority were married (93.1%). Moreover, 73.3% of the participants reported a monthly household income of less than 500 JOD, the average monthly household income in Jordan. The median Hba1c of the diabetic children was 9.0 (range 8.0–10.0), and their diabetes duration was 4.0 (range 2–5) years. Finally, only 6.4% reported vaccinating their children against the flu annually, and only 23% are planning to vaccinate their children this year.

Participants’ responses to the knowledge items regarding vaccinating their children against the flu are displayed in Table 2. The most frequently affirmed question was “Do you know how to properly use diabetes medications?” (93.6%), followed by the statement “The flu can spread from one person to another” (92.8%), while the most incorrectly answered question was “Can antibiotics can be used to treat the flu?” (58%). The median knowledge score was 8 (7–10) out of the maximum possible score of 12.

Participants’ attitudes and beliefs toward vaccinating their children against the flu varied. On the positive statements, most participants agreed/strongly agreed with the item “It is easy to reach the pharmacy /hospital to receive a flu vaccination” (58.2%), followed by the item “Flu vaccination prevents infection with the influenza virus” (50.4%), while the item participants least agreed/strongly agreed with was “My physician believes that my child should receive the flu vaccine” (31.6%). On the reverse-coded items, participants most disagreed/strongly disagreed with the item “Catching the flu is not a problem for my child” (55.6%), while the item participants least disagreed/strongly disagreed with was “The flu vaccination may cause complications/troubles for my child” (22%) (Table 3). The participants’ median attitude score towards vaccinating their children against the flu was 23 (range 20–27) out of the maximum possible score of 40.

Parents’ responses to the DM self-care practice items are provided in Table 4. On the positive statements, the items with the highest median were “On how many of the last SEVEN DAYS did you test your child’s blood sugar?”, “On how many of the last SEVEN DAYS did you test your child’s blood sugar the number of times recommended by your health care provider?”, and “On how many of the last SEVEN DAYS did you test your blood sugar?” with a median of 7 (range 5–7), while the item with the lowest median was 2 (range 1–5) for “On how many of the last SEVEN DAYS did your child participate in a specific exercise session (such as swimming, walking, biking)?”. The reversed-coded item was ” On how many of the last SEVEN DAYS did your child eat high-fat foods such as red meat or full-fat dairy products?” with a median of 2 (range 1–3). The participants’ score median was 4.3 (range 3.15–5) out of a maximum score of 7.

A multinomial regression model was employed to examine the relationship between various sociodemographic factors and the intention to vaccinate children against the flu, as demonstrated in Figure 1. The analysis revealed that as the age of the children increased, parents’ likelihood of refusing to vaccinate them against the flu decreased (OR = 0.846, 95% CI (0.736–0.974), *p* = 0.02). Additionally, as parents’ positive attitudes towards the flu vaccine increased, their odds of both rejection and hesitancy to vaccinate their children decreased (OR = 0.589, 95% CI (0.518–0.670), *p* < 0.001 and OR = 0.754, 95% CI (0.673–0.845), *p* < 0.001, respectively). Furthermore, participants who had never vaccinated their children against the flu had higher odds of refusing to vaccinate their children (OR = 2.515, 95% CI (1.015–6.235), *p* = 0.046). The model goodness of fit was confirmed by computing Nagelkerke’s R^2^, which was 0.61.

The reasons for parental refusal to vaccinate their children are displayed in Figure 2. The most common reason was “I think it might be harmful” (51.6%), followed by “I don’t think it is effective” (31.9%), while the least common reason was “Unavailability of the vaccine” (4.50%).

## 4. Discussion

The current study examined the acceptance of the flu vaccine among parents of diabetic children in Jordan, uncovering that factors such as knowledge, perceived risks, and healthcare system trust impacted parents’ decision to have their children vaccinated. The findings suggest a moderate acceptance rate, critically influenced by parents’ understanding of the heightened risks flu poses to diabetic children. 

The multinomial regression analysis revealed findings regarding the influence of children’s age and previous vaccination on parental decisions to vaccinate against the flu. Specifically, we observed that as the age of the children increased, parents were less likely to refuse vaccination. This could be attributed to a greater awareness or experience with the flu’s impact on older children, or a cumulative understanding of the vaccine’s benefits over time. Further research would be useful to explore this trend, in order to understand the dynamics of age-related vaccine acceptance among parents.

Additionally, the analysis highlighted that parents who had previously vaccinated their children against the flu were less likely to refuse vaccination in the current study. This prior positive experience with vaccination could reinforce trust in the vaccine’s efficacy and safety, reducing hesitancy in subsequent vaccination decisions. It highlights the importance of initial positive vaccination experiences in shaping long-term vaccination behaviors.

The relationship between parents’ positive attitudes towards the flu vaccine and their education level or healthcare recommendations was also a key finding. Parents with higher education levels and those who received strong recommendations from healthcare professionals were more likely to exhibit positive attitudes towards vaccination. This suggests that education and professional advice play crucial roles in shaping vaccine perceptions, pointing to the need for targeted communication strategies that make use of these influences to improve vaccine uptake.

In line with the present findings, prior research on parental attitudes towards childhood vaccinations indicates that sociodemographic factors, perceived risks and benefits, and the quality of information received play significant roles in vaccine acceptance [37]. Additionally, a study in the pediatric emergency department context revealed that parental attitudes significantly impact flu vaccine acceptance for their children [38]. 

The factors influencing Jordanian parents’ decision to vaccinate their children against the flu unearthed in our study are in line with similar research conducted in other countries. For instance, a US study highlighted that parents’ decisions to vaccinate their children are significantly influenced by their perceptions of the vaccine’s effectiveness and safety concerns, reflecting a critical need for clear and accessible vaccine information [39]. Moreover, as shown in the present study, a global systematic review showed that the attitudes of healthcare providers indirectly impact parental vaccine decisions, emphasizing the role of healthcare professionals in guiding public health initiatives [40]. Trust in healthcare professionals has been found to be associated with decreased flu vaccine hesitancy [41]. These findings underline the importance of addressing both cultural and systemic factors when developing flu vaccination campaigns and public health strategies aimed at parents of diabetic children in Jordan.

We identified several barriers to vaccination, including concerns over vaccine safety and accessibility challenges. These findings align with broader research in this area. A review of interventions synthesized parent-level barriers from systematic reviews, emphasizing common challenges such as misinformation and logistical hurdles, similar to those we observed in our Jordanian sample [42]. Furthermore, a study conducted in the United Kingdom identified specific barriers such as mistrust in vaccine efficacy and healthcare systems, which mirrors the skepticism we noted in our study participants [43]. It has been suggested that educational approaches could address the concerns highlighted in our study, such as improving parents’ vaccine literacy and access to the vaccine [44]. A US study showed that educational interventions, such as videos or infographics, significantly improved patients’ knowledge about COVID-19 and vaccines, leading to increased vaccine acceptance. This demonstrates the potential impact of educational tools in enhancing vaccine literacy and addressing concerns about vaccine safety and efficacy [45]. Furthermore, policy-driven strategies may be useful; for example, mandating healthcare providers to engage in conversations with parents about the flu vaccine during routine visits would help ensure that parents of diabetic children receive consistent and accurate information, fostering an environment of trust and informed decision-making. Including the flu vaccine in national immunization programs and subsidizing the cost for low-income families are additional policy-driven approaches that may improve access and uptake, making vaccines more accessible to a broader segment of the population.

### Strengths, Limitations, and Future Directions

Our study has several strengths that contribute to the existing body of knowledge on vaccine acceptance. Firstly, the methodology allowed for in-depth analysis of parental attitudes, providing important insights into the specific concerns and motivations influencing vaccination decisions. Additionally, our focus on Jordan offers valuable context-specific findings that can inform local public health strategies to increase flu vaccine uptake in diabetic children. Finally, the Nagelkerke’s R^2^ of the logistic regression model was 0.61 indicating that approximately as much as 61% of the variance of the outcomes (parental intention to vaccinate their diabetic children in the current year) was accounted for by the model, confirming model fitness and that the predictions produced are highly reliable.

On the other hand, the reliance on self-reported data and the confined geographic scope of the study impact the generalizability of our findings. The reliance on self-reported data may have introduced response bias, as participants may have provided socially desirable answers, or their recollections may have been inaccurate. To mitigate this, we ensured confidentiality, aiming to encourage honest responses. The study’s confined geographic scope, focusing on two public Jordanian hospitals, may limit the generalizability of our findings to the entire Jordanian population or other cultural contexts. Future studies could broaden the geographic scope and include a more diverse sample to enhance representativeness.

Moreover, the cross-sectional design of our study captures attitudes and acceptance at a single point in time, which may not reflect changes in perceptions due to evolving healthcare policies, public health campaigns, or disease outbreaks. Longitudinal studies could provide more dynamic insights into how parental attitudes towards vaccination evolve over time.

Another potential limitation is the selection bias inherent in our sampling method. Parents who agreed to participate might inherently have different attitudes towards healthcare and vaccination compared to non-participants. To address this, we attempted to approach a representative sample of parents visiting the selected clinics, ensuring a broad spectrum of sociodemographic backgrounds.

As this was a quantitative study, the depth of information obtained may be limited. Using qualitative methods in future work could provide a more detailed point of view of the participants on specific topics. Future research should also explore the longitudinal effects of educational interventions and the efficacy of various communication strategies in improving flu vaccine uptake in this population. 

## 5. Conclusions

Our study sheds light on the factors influencing flu vaccine acceptance among parents of diabetic children in Jordan. It indicates that misconceptions about vaccine safety and efficacy are significant barriers to flu vaccine acceptance among Jordanian parents of diabetic children. The findings of the multinomial regression analysis suggest that addressing these misconceptions is crucial for improving vaccine uptake. Future efforts should focus on enhancing healthcare communication and educational strategies to mitigate these barriers and increase vaccination rates in this vulnerable population group. This calls for a comprehensive approach, integrating education to address misinformation, healthcare provider engagement to harness their influence on parents, and policy reform to bolster vaccine uptake. Addressing the identified barriers through comprehensive public health strategies could significantly advance the protection of Jordanian diabetic children against the flu.

## Figures and Tables

**Figure 1 vaccines-12-00262-f001:**
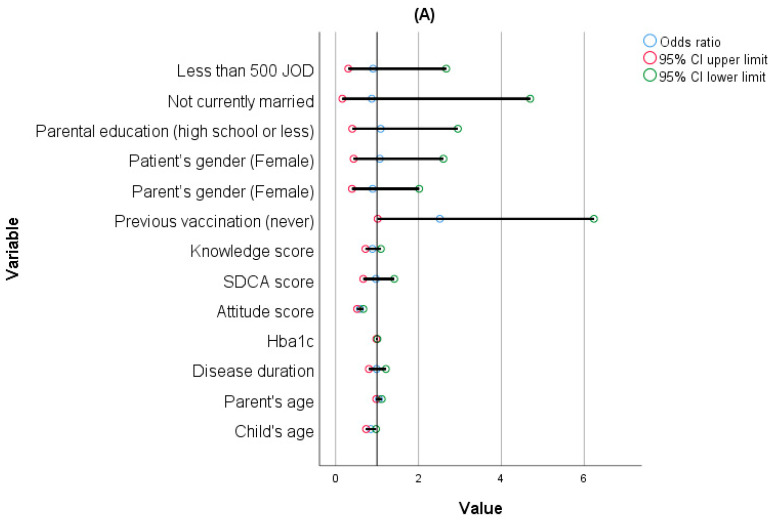
Multinomial regression analysis of sociodemographic factors and parental intentions regarding vaccination of their children against flu: (**A**) Yes vs. No, (**B**) Yes vs. Not sure.

**Figure 2 vaccines-12-00262-f002:**
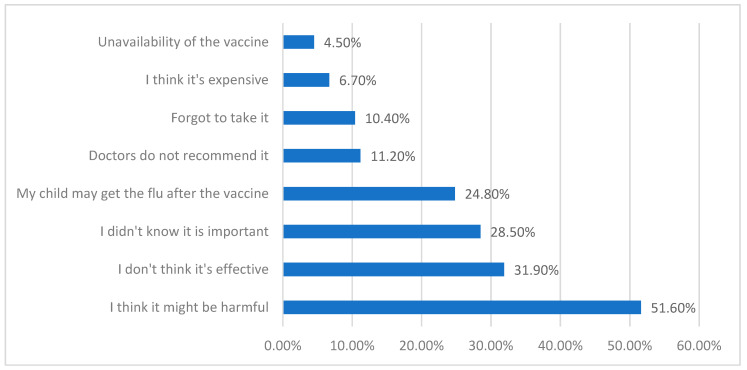
Reasons for parents’ refusal to vaccinate their diabetic children against the flu.

**Table 1 vaccines-12-00262-t001:** Demographic profile of the study participants.

		Median (25–75) Or Frequency (%)
Child’s age		10.0 (6.0–13.0)
Parent’s age		37 (32–43)
Child’s gender	Female	210 (51.9%)
	Male	195 (48.1%)
Parent’s gender	Female	253 (62.5%)
	Male	152 (37.5%)
Level of education	High school or lower	244 (60.2%)
	University/college degree	161 (39.8%)
Social status	Other	28 (6.9%)
	Married	377 (93.1%)
Income (Jordanian Dinars) *	Less than 500	297 (73.3%)
	500 or more	108 (26.7%)
Hba1c		9.0 (8.0–10.0)
Disease duration		4.0 (2.0–5.0)
Intent to vaccinate your child this year?	Yes	93 (23%)
	Not sure	118 (29.1)
	No	194 (47.9)
Previous flu vaccination	Never	277 (68.4)
	One	55 (13.5%)
	More than one	47 (11.6%)
	Annually	26 (6.4%)

* 500 Jordanian Dinar = 705 USD.

**Table 2 vaccines-12-00262-t002:** Participants’ knowledge regarding vaccinating their children against the flu.

	No	Unsure	Yes
Do you know how to measure your child’s blood sugar levels at home? *	33 (8.1%)	8 (2%)	364 (89.9%)
Are you aware that weight loss can be a sign of diabetes? *	65 (16%)	100 (24.7%)	240 (59.3%)
Is diabetes hereditary? *	167 (41.2%)	21 (5.2%)	217 (53.6%)
Is diabetes a chronic disease? *	36 (8.9%)	9 (2.2%)	360 (88.9%)
Can you recognize the symptoms of low blood sugar? *	48 (11.9%)	9 (2.2%)	348 (85.9%)
Is there a vaccine against the flu? *	35 (8.6%)	125 (30.9%)	245 (60.5%)
Does the vaccine have side effects? *	35 (8.6%)	249 (61.5%)	121 (29.9%)
The flu is caused by bacteria **	207 (51.1%)	86 (21.2%)	112 (27.7%)
The flu can spread from one person to another. *	18 (4.4%)	11 (2.7%)	376 (92.8%)
Do you know how to properly use diabetes medications? *	22 (5.4%)	4 (1.0%)	397 (93.6%)
Can antibiotics be used to treat the flu? **	146 (36%)	24 (5.9%)	235 (58.0%)
When is the appropriate time to take the flu vaccine? #			
January–March	September–October	November–December	Unsure
16 (4.0%)	114 (28.1%)	42 (10.4%)	233 (57.5%)

* “yes” is the correct answer. ** “no” is the correct answer. # “September–October “is the correct answer.

**Table 3 vaccines-12-00262-t003:** Frequencies (%) of participants’ attitudes toward the flu vaccine.

	Strongly Disagree	Disagree	Neutral	Agree/Strongly Agree
I believe that my child must receive the flu vaccination	29 (7.2%)	130 (32.1%)	77 (19%)	169 (41.7%)
It is easy to reach the pharmacy/hospital to receive a flu vaccination	49 (12.1%)	60 (14.8%)	60 (14.8%)	236 (58.2%)
My physician believes that my child should receive the flu vaccine	25 (6.2%)	85 (21.1%)	165 (41%)	127 (31.6%)
Flu vaccination prevents infection by the influenza virus	8 (2%)	76 (18.9%)	115 (28.6%)	203 (50.4%)
The flu vaccination may cause complications/troubles for my child *	19 (4.7%)	70 (17.3%)	100 (24.8%)	215 (53.2%)
I believe that my child gets sick because of the flu shot *	19 (4.7%)	85 (21%)	105 (25.9%)	196 (48.4%)
I am worried about the chances of my child contracting the flu because of the flu vaccine *	20 (5%)	84 (20.9%)	77 (19.2%)	221 (55%)
Catching the flu is not a problem for my child *	75 (18.6%)	149 (37%)	35 (8.7%)	144 (35.7%)

* Reversed coded statement.

**Table 4 vaccines-12-00262-t004:** Parents’ responses to diabetes self-care practices of their diabetic children.

	Median	Percentile 25	Percentile 75
How many of the last SEVEN DAYS has your child followed a healthful eating plan?	5	3	7
On average, over the past month, how many DAYS PER WEEK has your child followed his/her eating plan?	4	3	7
On how many of the last SEVEN DAYS did your child eat five or more servings of fruits and vegetables?	3	2	7
On how many of the last SEVEN DAYS did your child eat high-fat foods such as red meat or full-fat dairy products? *	2	1	3
On how many of the last SEVEN DAYS did your child participate in at least 30 min of physical activity? (Total minutes of continuous activity, including walking).	5	3	6
On how many of the last SEVEN DAYS did your child participate in a specific exercise session (such as swimming, walking, biking)	2	1	5
On how many of the last SEVEN DAYS did you test your child’s blood sugar?	7	5	7
On how many of the last SEVEN DAYS did you test your child’s blood sugar the number of times recommended by your health care provider?	7	5	7
On how many of the last SEVEN DAYS did you check your child’s feet?	5	2	7
On how many of the last SEVEN DAYS did you inspect the inside of your child’s shoes?	5	1	7

* Reverse-coded item.

## Data Availability

The data presented in this study are openly available in [Zenodo] at [10.5281/zenodo.10728606], reference number [46].

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
