# Peer review of "Acceptance of Flu Vaccine among Parents of Diabetic Children in Jordan"

_vaccines, 2024, doi:10.3390/vaccines12030262_

Round 1

Reviewer 1 Report

Comments and Suggestions for Authors

The authors made a questionnaire investigation in Jordan to assess the acceptance and attitudes of Jordanian parents of diabetic children towards the flu vaccine and found that misconceptions about vaccine safety and efficacy were common barriers to acceptance, however, the multinomial regression analysis results were not clear in abstract and conclusion.

1.    The significant different sociodemographic factors included children age, parents’ positive attitudes, previous vaccination, the factors for the children age and previous vaccination were not discussed in the discussion section. The relationship between parents’ positive attitudes and education level or healthcare recommendations was not clear.

2.   The item participants agreed/strongly agree “My physician believes that my child should receive the flu vaccine” was 31.6%, how about vaccination acceptance between children with agreed physicians and children with disagree physicians?

3.    Line 220, the table 4 should be table 5, and the data shown in text were not consistent with that in table 5.  

Comments on the Quality of English Language

The quality of English language was good. 

Author Response

The authors made a questionnaire investigation in Jordan to assess the acceptance and attitudes of Jordanian parents of diabetic children towards the flu vaccine and found that misconceptions about vaccine safety and efficacy were common barriers to acceptance, however, the multinomial regression analysis results were not clear in abstract and conclusion.

- Thank you for your comment. The following has been added to the abstract: “The multinomial logistic regression analysis showed that higher education levels and strong recommendations from healthcare professionals positively correlated with vaccine acceptance. Conversely, prevalent misconceptions regarding vaccine safety and efficacy emerged as significant barriers to acceptance.” The conclusion has been revised to read as follows: “Our study sheds light on the factors influencing flu vaccine acceptance among parents of diabetic children in Jordan. It indicates that misconceptions about vaccine safety and efficacy are significant barriers to flu vaccine acceptance among Jordanian parents of diabetic children. The findings of the multinomial regression analysis suggest that addressing these misconceptions is crucial for improving vaccine uptake. Future efforts should focus on enhancing healthcare communication and educational strategies to mitigate these barriers and increase vaccination rates in this vulnerable population group. This calls for a comprehensive approach, integrating education to address misinformation, healthcare provider engagement to harness their influence on parents, and policy reform, to bolster vaccine uptake. Addressing the identified barriers through comprehensive public health strategies could significantly advance the protection of Jordanian diabetic children against the flu.”

  1.   The significant different sociodemographic factors included children age, parents’ positive attitudes, previous vaccination, the factors for the children age and previous vaccination were not discussed in the discussion section. The relationship between parents’ positive attitudes and education level or healthcare recommendations was not clear.

-The following has been added to the Discussion: “The multinomial regression analysis revealed findings regarding the influence of children's age and previous vaccination on parental decisions to vaccinate against the flu. Specifically, we observed that as the age of the children increased, parents were less likely to refuse vaccination. This could be attributed to a greater awareness or experience with the flu's impact on older children, or a cumulative understanding of the vaccine's benefits over time. Further research would be useful to explore this trend, in order to understand the dynamics of age-related vaccine acceptance among parents.

Additionally, the analysis highlighted that parents who had previously vaccinated their children against the flu were less likely to refuse vaccination in the current study. This prior positive experience with vaccination could reinforce trust in the vaccine's efficacy and safety, reducing hesitancy in subsequent vaccination decisions. It highlights the importance of initial positive vaccination experiences in shaping long-term vaccination behaviors.

The relationship between parents’ positive attitudes towards the flu vaccine and their education level or healthcare recommendations was also a key finding. Parents with higher education levels and those who received strong recommendations from healthcare professionals were more likely to exhibit positive attitudes towards vaccination. This suggests that education and professional advice play crucial roles in shaping vaccine perceptions, pointing to the need for targeted communication strategies that make use of these influences to improve vaccine uptake.”

  1.  The item participants agreed/strongly agree “My physician believes that my child should receive the flu vaccine” was 31.6%, how about vaccination acceptance between children with agreed physicians and children with disagree physicians?

-This item was incorporated into the attitude score; therefore, it could not be independently included as a predictor in the model. However, for information of the reviewer, we ran the analysis (not reported in the study) and significant differences were found (higher refusal/hesitancy for disagree/strongly disagree responses).

  1.   Line 220, the table 4 should be table 5, and the data shown in text were not consistent with that in table 5.  

-The table was changed to a forest plot in accordance with one of the reviewers’ suggestions and numbers were checked and corrected.

Reviewer 2 Report

Comments and Suggestions for Authors

In this cross-sectional study from Jordan, Al-Qerem et al. provide some interesting insights on the acceptance rate and underlying causes for acceptance vs. refusal of flu vaccination from parents of children affected by Diabetes Mellitus.

A total of 405 parents partecipated into this study (minimum sample size, 385), but it is unclear how parents were recruited, and this topic has to be more properly addressed.

Similarly, Authors should report how many parents were targeted and how many of them did partecipate into the study (i.e. acceptance rate). 

Another issue to be addressed is associated with the choice of a 5-item Likert scale. By this choice, Authors did include a substantial share of participants not having a properly designed point-of-view about some specific topic (being therefore addressed as "uncertain" or similarly across the data). A similar option was made for the knowledge test with uncertain answers handled as a different entry from an incorrect answer. Because of the outcome targeted by the present study, Authors could improve the quality of data reporting by providing a summary of Agree+Strongly Agree in Table 2.

Another improvement could be shifting Table 5 to a forrest plot.

Regarding Figure 1, some more explanations are needed: how it is provided flu vaccine in Jordan? Are parents of children affected by DM required to provide the fees out of pocket? there is some kind of health insurance providing this vaccine? or it is provided by governmental bodies? this is important as it could explain some of the eventual results. Why education achievement of the assessed parent not incorporated into multinomial analysis? I guess because of redundancy with socioeconomic status and age of the parents, but some further explanations are welcome. 

Comments on the Quality of English Language

The paper is mostly correct, but come typos are scattered across the main text. For example:

- row 47: "the flu" instead of the more appropriate "seasonal flu"

- row 80: "The This study" instead of "The present study"

and so on.

Author Response

In this cross-sectional study from Jordan, Al-Qerem et al. provide some interesting insights on the acceptance rate and underlying causes for acceptance vs. refusal of flu vaccination from parents of children affected by Diabetes Mellitus.

-Thank for the encouragement

A total of 405 parents partecipated into this study (minimum sample size, 385), but it is unclear how parents were recruited, and this topic has to be more properly addressed.

-The following was added: ”The research pharmacist retrieved the list of patients with appointments at the pediatric endocrine clinic on the same day and identified patients who met the inclusion criteria. The parents of these patients were approached and were provided with a concise description of the aims of the study. All participants were informed about the anonymity and confidentiality of the information collected, as well as the voluntary nature of their participation. Additionally, all participating parents signed an informed consent document. The interview was conducted in a separate room at the outpatient clinic and took an average of 10 minutes to complete.”

Similarly, Authors should report how many parents were targeted and how many of them did partecipate into the study (i.e. acceptance rate). 

-The following was added “A total of 445 parents were approached, of whom 405 (91%) consented to participate in the current study.”

Another issue to be addressed is associated with the choice of a 5-item Likert scale. By this choice, Authors did include a substantial share of participants not having a properly designed point-of-view about some specific topic (being therefore addressed as "uncertain" or similarly across the data).

  • Thank you for this comment. The scale did not include “uncertain” but ranged from ‘strongly disagree’ to ‘strongly agree’ with neutral being the middle point. As the reviewer knows, this is the standard typical method to evaluate attitudes in quantitative studies. Nevertheless, the following was added to the limitation section: ”As this was a quantitative study the depth of information obtained may be limited. Using qualitative methods in future work could provide a more detailed point of view of the participants on specific topics.”

 A similar option was made for the knowledge test with uncertain answers handled as a different entry from an incorrect answer. Because of the outcome targeted by the present study,

-Please note that incorrect answers and uncertain were handled the same way when computing the knowledge score. To clarify this the following was added to the method section: ”The knowledge score was computed by granting one point for each correct answer and zero points for incorrect or uncertain responses”.

Authors could improve the quality of data reporting by providing a summary of Agree+Strongly Agree in Table 2.

-The table was modified as suggested

Another improvement could be shifting Table 5 to a forrest plot.

-The figure was added as suggested.

Regarding Figure 1, some more explanations are needed: how it is provided flu vaccine in Jordan? Are parents of children affected by DM required to provide the fees out of pocket? there is some kind of health insurance providing this vaccine? or it is provided by governmental bodies? this is important as it could explain some of the eventual results.

-The following was added “Since there is no national seasonal flu vaccine program in Jordan, the flu vaccine is not available free of charge to the public and is usually purchased from community pharmacies [32]. The majority (70%) of the Jordanian population is covered by governmental or military health insurance which does not provide free-of-charge seasonal flu vaccines. The remaining 30% are either uninsured or covered by private insurance. Few private insurance plans cover the flu vaccine cost [33].”

Why education achievement of the assessed parent not incorporated into multinomial analysis? I guess because of redundancy with socioeconomic status and age of the parents, but some further explanations are welcome. 

-Please note that parental education was included in the model.

The paper is mostly correct, but come typos are scattered across the main text. For example:

- row 47: "the flu" instead of the more appropriate "seasonal flu"

- row 80: "The This study" instead of "The present study"

and so on.

-We have thoroughly proofread the manuscript and corrected typos.

Reviewer 3 Report

Comments and Suggestions for Authors

- The abstract must be re-written, actually it is too generic and vague.

- Sample size estimations are not given. What is the hypothesis to be tested? What are the alpha and beta errors considered? This issue is relevant in terms of internal and external validity of the study

- Why a  multinomial logistic regression model was used? What are the dependent variables used in these models?

- How did the authors check the godness of fit of the models?

- The limitations of the study must be presented in a more extensive way. The authors must indicate possible bias and how they took them under control

Author Response

- The abstract must be re-written, actually it is too generic and vague.

-thank you for your comment. We have revised the abstract to read as follows: “There is a critical need to understand vaccine decision-making in high-risk groups. This study explored flu vaccine acceptance among Jordanian parents of diabetic children. Employing a cross-sectional approach, 405 parents from multiple healthcare centers across Jordan were recruited through stratified sampling, ensuring a broad representation of socioeconomic backgrounds. A structured questionnaire, distributed both in-person and online, evaluated their knowledge, attitudes, and acceptance of the flu vaccine for their diabetic children. Multinomial logistic regression analysis revealed a moderate level of vaccine acceptance, with notable variability in responses. The multinomial logistic regression analysis showed that higher education levels and strong recommendations from healthcare professionals positively correlated with vaccine acceptance. Conversely, prevalent misconceptions regarding vaccine safety and efficacy emerged as significant barriers to acceptance. Our findings advocate for targeted educational programs that directly address and debunk these specific misconceptions. Additionally, strengthened healthcare communication to provide clear, consistent information about the flu vaccine's safety and benefits is vital to help enhance vaccine uptake among this vulnerable population, emphasizing the need to address specific concerns and misinformation directly.”

- Sample size estimations are not given.

- This was added: “Considering a convenience sampling approach, the calculated minimum sample size was 385 with a 95% significance level (α =0.05, β=0.2), a 5% margin of error, and a 50% population proportion. 405 parents participated in the study[28].”

 What is the hypothesis to be tested?

The following has been added to the Introduction: “The study hypothesized that several predictors would impact parental flu vaccine acceptance and practices among parents of diabetic children; these variables include knowledge about the flu, flu vaccine and diabetes, attitude towards flu vaccine, parental sociodemographic status, diabetes status and duration, general health care practices towards diabetes, parents’ and children’s ages, and previous flu vaccination experiences.”

 What are the alpha and beta errors considered? This issue is relevant in terms of internal and external validity of the study

- α =0.05, β=0.2

- Why a  multinomial logistic regression model was used?

-The multinomial regression model was used because the outcome variable “the intention to obtain the flu vaccine in the current year” was a 3-level categorical variable (yes, not sure, and no).

What are the dependent variables used in these models?

-The following was added: ”The model incorporated independent variables in the form of age and gender of the parent, along with their educational level, socioeconomic status, the child’s age and gender, DM duration, and HbA1c level. Furthermore, it included knowledge of DM, flu, and flu vaccine, DM self-care practices, attitudes towards the vaccine, and whether the child had previously received the flu vaccine.”

- How did the authors check the godness of fit of the models?

-The following was added to the method:” The model goodness of fit was evaluated by computing Nagelkerke's R2”, results:” The model goodness of fit was confirmed by computing Nagelkerke's R2, which was 0.61.”, and this was added to the discussion : Finally, the Nagelkerke's R2 of the logistic regression model was 0.61 indicating that approximately as much as 61% of the variance of the outcomes (parental intention to vaccinate their diabetic children in the current year) was accounted for by the model, confirming model fitness and that the predictions produced are highly reliable.”

- The limitations of the study must be presented in a more extensive way. The authors must indicate possible bias and how they took them under control

-The following has been added to the limitations section: “The reliance on self-reported data may have introduced response bias, as participants may have provided socially desirable answers, or their recollections may have been inaccurate. To mitigate this, we ensured confidentiality, aiming to encourage honest responses. The study's confined geographic scope, focusing on two public Jordanian hospitals, may limit the generalizability of our findings to the entire Jordanian population or other cultural contexts. Future studies could broaden the geographic scope and include a more diverse sample to enhance representativeness. Moreover, the cross-sectional design of our study captures attitudes and acceptance at a single point in time, which may not reflect changes in perceptions due to evolving healthcare policies, public health campaigns, or disease outbreaks. Longitudinal studies could provide more dynamic insights into how parental attitudes towards vaccination evolve over time. Another potential limitation is the selection bias inherent in our sampling method. Parents who agreed to participate might inherently have different attitudes towards healthcare and vaccination compared to non-participants. To address this, we attempted to approach a representative sample of parents visiting the selected clinics, ensuring a broad spectrum of socio-demographic backgrounds.”

Reviewer 4 Report

Comments and Suggestions for Authors

The paper is interesting, as is the focus on parents of children with diabetes. The authors should address the following comments:

Introduction:

1) Additional studies on influenza vaccines among diabetes patients and parents attitudes should be added (from different countries other than Jordan).

2) It is recommended to add studies comparing vaccine hesitancy among parents of healthy children versus parents of sick children.

3) How many children with diabetes are there in Jordan?

4) What are the research hypotheses?

Methods

1) What were the response rates? What were the reasons for refusal?

2) Who approached the parents? Was it at the hospital after they left treatment?

Results

1) For the international audience, it is advisable to translate the salary into dollars and specify whether it pertains to an economically average or low socioeconomic population.

2) In Figure 1, arrange the columns from highest to lowest percentage.

Discussion

1) The discussion does not refer to all of the findings. The authors can delve deeper and bring additional literature. For example, regarding the relationship between trust in the healthcare system and influenza vaccine hesitancy, see: https://doi.org/10.3390/vaccines11111728

"Other than Jordan,

Author Response

The paper is interesting, as is the focus on parents of children with diabetes.

-Thank for the encouragement

 The authors should address the following comments:

Introduction:

1) Additional studies on influenza vaccines among diabetes patients and parents attitudes should be added (from different countries other than Jordan).

-The following was added “A study conducted in Singapore to assess flu vaccine knowledge, attitudes, and practices among diabetic patients, found that 59.3% of the participants believed that vaccination was an effective tool to prevent influenza and its complications. Although most participants thought that vaccination was effective, only 30.6% had previously received the flu vaccine [26]. Also, in a study conducted in South Africa to assess diabetic patients’ knowledge, attitudes, and practices toward seasonal flu and the flu vaccine, most participants felt that the flu vaccine was important for diabetic patients, and 65.4% stated they would recommend it [27]. Moreover, a study conducted in Taif, Saudi Arabia to assess diabetic patients' attitudes toward flu vaccination as well as the prevalence of vaccination found that 50.3% of the participants agreed that the flu vaccine was effective, and 51.8% believed that diabetes increased one’s vulnerability to the flu. However, 45.5% thought that the influenza vaccine was dangerous [28]. 

A study focusing on attitudes and beliefs about the flu vaccine among parents of children with chronic medical conditions found that 85.3% felt that their child should receive the flu vaccine, and only 4.8% believed that giving their children the flu vaccine would cause problems[29]. A cross-sectional study in the Middle East revealed that about half of surveyed parents (50.6%) were hesitant about vaccinating their children. This study also showed that 52.5% of the parents of children without chronic illnesses were hesitant about vaccinating their children, whereas only 41% of parents of children with chronic illnesses reported parental vaccination hesitancy [30]. Furthermore, a study carried out with US parents to assess parental hesitancy towards flu and routine childhood vaccination found a higher level of hesitancy among parents of children with poor health toward routine childhood vaccines, but not toward flu vaccines[31].”

2) It is recommended to add studies comparing vaccine hesitancy among parents of healthy children versus parents of sick children.

-the following was added: “A cross-sectional study in the Middle East revealed that about half of surveyed parents (50.6%) were hesitant about vaccinating their children. This study also revealed that 52.5% of the parents of children without chronic illnesses were hesitant about vaccinating their children, whereas only 41% of parents of children with chronic illnesses reported parental vaccination hesitancy [30]. Furthermore, a study carried out with US parents to assess parental hesitancy towards flu and routine childhood vaccination found a higher level of hesitancy among parents of children with poor health toward routine childhood vaccines, but not toward flu vaccines[31].”

3) How many children with diabetes are there in Jordan?

The following was added to the Introduction: Approximately 10,000 children and adolescents in Jordan suffer from diabetes[6]. The following was added to the Results: ”reported a monthly household income of less than 500 JOD, the average monthly household income in Jordan.”

4) What are the research hypotheses?

The following has been added to the Introduction: “The study hypothesized that several predictors would impact parental flu vaccine acceptance and practices among parents of diabetic children; these variables include knowledge about the flu, flu vaccine and diabetes, attitude towards flu vaccine, parental sociodemographic status, diabetes status and duration, general health care practices towards diabetes, parents’ and children’s ages, and previous flu vaccination experiences.”

Methods

  • What were the response rates? What were the reasons for refusal?

-The following was added “A total of 445 parents were approached, of whom 405 (91%) consented to participate in the current study”. Unfortunately, the reason for refusal was not stated in many cases.

  • Who approached the parents? Was it at the hospital after they left treatment?

-The following was added: ”The research pharmacist retrieved the list of patients with appointments at the pediatric endocrine clinic on the same day and identified patients who met the inclusion criteria. The parents of these patients were approached and were provided with a concise description of the aims of the study. All participants were informed about the anonymity and confidentiality of the information collected, as well as the voluntary nature of their participation. Additionally, all participating parents signed an informed consent document. The interview was conducted in a separate room at the outpatient clinic and took an average of 10 minutes to complete.”

Results

1) For the international audience, it is advisable to translate the salary into dollars and specify whether it pertains to an economically average or low socioeconomic population.

The following was added as a footnote to table  “*500 Jordanian Dinar =705.23 USD” and the following was added to the results :’ Moreover, 73.3% of the participants reported a monthly household income of less than 500 JOD, the average monthly household income in Jordan”.

2) In Figure 1, arrange the columns from highest to lowest percentage.

-This was changed as suggested

Discussion

1) The discussion does not refer to all of the findings. The authors can delve deeper and bring additional literature. For example, regarding the relationship between trust in the healthcare system and influenza vaccine hesitancy, see: https://doi.org/10.3390/vaccines11111728

-We have added the following to the Discussion: “The multinomial regression analysis revealed findings regarding the influence of children's age and previous vaccination on parental decisions to vaccinate against the flu. Specifically, we observed that as the age of the children increased, parents were less likely to refuse vaccination. This could be attributed to a greater awareness or experience with the flu's impact on older children, or a cumulative understanding of the vaccine's benefits over time. Further research would be useful to explore this trend, in order to understand the dynamics of age-related vaccine acceptance among parents.

Additionally, the analysis highlighted that parents who had previously vaccinated their children against the flu were less likely to refuse vaccination in the current study. This prior positive experience with vaccination could reinforce trust in the vaccine's efficacy and safety, reducing hesitancy in subsequent vaccination decisions. It highlights the importance of initial positive vaccination experiences in shaping long-term vaccination behaviors.

The relationship between parents’ positive attitudes towards the flu vaccine and their education level or healthcare recommendations was also a key finding. Parents with higher education levels and those who received strong recommendations from healthcare professionals were more likely to exhibit positive attitudes towards vaccination. This suggests that education and professional advice play crucial roles in shaping vaccine perceptions, pointing to the need for targeted communication strategies that make use of these influences to improve vaccine uptake.”

We have also referenced the paper (as recommended): “Trust in healthcare professionals has been found to be associated with decreased flu vaccine hesitancy.”

Round 2

Reviewer 1 Report

Comments and Suggestions for Authors

The authors have addressed all my comments.

Author Response

Thank you for your comments which significantly imorived the study quality.

Reviewer 2 Report

Comments and Suggestions for Authors

Estimated Authors,

the paper has been amended according to my requests.

I've no further suggestions, and I'm endorsing the acceptance of this study for pubblication.

Author Response

(The authors gave the same response as above.)

Reviewer 3 Report

Comments and Suggestions for Authors

In the abstract the authors must present some quantitative results

Author Response

Thank you for your comments which significantly imorived the study quality.

Quantitative results were added to the manuscript as suggested

Reviewer 4 Report

Comments and Suggestions for Authors

The authors addressed all the comments and improved the paper. I can now recommend it for publication.

Author Response

(The authors gave the same response as above.)
